# Actionable Molecular Alterations Are Revealed in Majority of Advanced Non-Small Cell Lung Cancer Patients by Genomic Tumor Profiling at Progression after First Line Treatment

**DOI:** 10.3390/cancers14010132

**Published:** 2021-12-28

**Authors:** Malene Støchkel Frank, Uffe Bodtger, Julie Gehl, Lise Barlebo Ahlborn

**Affiliations:** 1Department of Clinical Oncology and Palliative Care, Zealand University Hospital, 4000 Roskilde, Denmark; kgeh@regionsjaelland.dk; 2Department of Clinical Medicine, Faculty of Health and Medical Sciences, University of Copenhagen, 2200 Copenhagen, Denmark; 3Department of Respiratory Medicine, Zealand University Hospital, 4700 Naestved, Denmark; ubt@regionsjaelland.dk; 4Institute for Regional Health Research, University of Southern Denmark, 5000 Odense, Denmark; 5Center for Genomic Medicine, Rigshospitalet, Copenhagen University Hospital, 2100 Copenhagen, Denmark; Lise.Barlebo.Ahlborn@regionh.dk

**Keywords:** NSCLC, re-biopsy, genomic profiling, resistance mechanisms, therapeutic pressure, targeted treatment, precision medicine

## Abstract

**Simple Summary:**

In precision medicine, cancer patients are treated with drugs that target specific molecular alterations found in their cancer cells. As new drugs for specific targets emerge, possibilities expand. Our prospective clinical study explored the possibilities of receiving targeted treatments after standard first line treatment by performing genomic profiling of a biopsy taken at diagnosis, as well as at time of progression, in patients with advanced non-small cell lung cancer. In the majority of patients (85%), there was a potential of receiving targeted treatment, based on the re-biopsy results. In approximately one third of patients, we found new molecular alterations not present at the diagnostic biopsy, strengthening the relevance of performing a re-biopsy at progression to increase targeted treatment options, and hopefully bettering the prognosis.

**Abstract:**

*Background*: Genomic profiling in advanced Non-Small Cell Lung cancer (NSCLC) can reveal Actionable Molecular Alterations (AMAs). Our study aims to investigate clinical relevance of re-biopsy after first line treatment, by reporting on acquired and persistent AMAs and potential targeted treatments in a real-time cohort of NSCLC patients. *Methods*: Patients with advanced NSCLC receiving first-line treatment were prospectively included in an observational study (NCT03512847). Genomic profiling was performed by TruSight Oncology 500 HT gene panel on tumor tissue collected at diagnosis and at time of progression. *Results:* The 92 patients re-biopsied at progression had received immunotherapy (*n* = 44), chemotherapy (*n* = 44), or combination treatment (*n* = 4). In 87 of these patients (95%), successful genomic profiling was performed at both the diagnostic biopsy and the re-biopsy. In 74 patients (85%), ≥1 AMA were found. The AMAs were acquired in 28%. The most frequent AMAs were observed in *TP53* (45%), *KRAS* (24%), *PIK3CA* (6%), and *FGFR1* (6%). Only five patients (5%) received targeted treatment mainly due to deterioration in performance status. *Conclusions*: Re-biopsy at progression revealed acquired AMAs in approximately one third of patients, and 85% had at least one AMA with the potential of receiving targeted treatment, thus strengthening the clinical relevance of re-biopsy.

## 1. Introduction

An increasing number of targeted treatments has been introduced in the field of precision medicine over the last decades. Particularly in advanced Non-Small Cell Lung Cancer (NSCLC), treatment options have evolved with epidermal growth factor receptor (*EGFR*), anaplastic lymphoma kinase (*ALK*)- and *ROS* proto-oncogene 1 (*ROS1*) directed treatment [1,2,3]. Despite the increasing eligibility and response to genome-targeted therapy from 2006 (3%) to 2020 (7%) [4], the clinical benefit of precision medicine, beyond standard of care treatments, is still described as uncertain [5]. Real-time studies in advanced NSCLC have not yet demonstrated any improvements in survival by comprehensive molecular profiling [6,7,8], but this scenario might change due to an increasing numbers of promising targeted treatments.

Reports from precision medicine clinical trials have shown that in 40–75% of patients, a potential targeted treatment is available, but only 8–27% of patients actually receive targeted treatment [9,10,11,12,13,14,15,16,17,18,19], mainly due to patients’ deterioration. It is currently debated if comprehensive molecular profiling should be performed before exhaustion of treatment options to increase the benefit of targeted treatment. In addition, it is questioned if a re-biopsy is required, with the potential risk of complications, or if the diagnostic samples provide adequate molecular information in treatment-guidance [20].

To our knowledge, only few prospective studies have investigated the concordance between a diagnostic biopsy and re-biopsy in the metastatic setting, and these are mainly reporting from heterogeneous cohorts of patients with a variety of different cancer types receiving different kinds of treatments [21,22]. A recent review (2019) concluded that in advanced NSCLC, the significance of a re-biopsy is highly dependent on the clinical situation [20]. By the increasing knowledge of development of new molecular alterations during tumor evolution and potential resistance mechanisms [23] it is indicated that real-time comprehensive molecular profiling is preferred. Altogether, this underlines the importance of prospective real-time studies in homogenous patient cohorts with performance of re-biopsy at specific time-points to state the clinical relevance of re-biopsy in terms of revealing acquired actionable molecular alterations with the potential of targeted treatment.

Our study reports a real-time cohort of 150 advanced or non-curative locally advanced NSCLC patients, having the availability of a re-biopsy on progression after first line treatment, and the results strengthens the clinical relevance of re-biopsy.

## 2. Materials and Methods

### 2.1. Study Design

The study is a prospective, explorative single-center study (ClinicalTrials.gov NCT03512847) conducted at Department of Clinical Oncology and Palliative Care, Zealand University Hospital (Næstved Hospital). It was approved by The Regional Committee on Health Research Ethics (SJ-662) and The Danish Data Protection Agency (REG-006-2018), and was conducted according to the Helsinki Declaration. All patients provided signed informed consent.

This cohort was described in a previous publication detailing the feasibility of re-biopsy and complication rate thereof, as well as changes in Programmed Death Ligand 1 (PD-L1) expression [24].

The primary aim was to investigate the clinical relevance of re-biopsy after first line treatment in advanced NSCLC patients by reporting on actionable molecular alterations and potential targeted treatments. In addition, we investigated if the molecular alterations were acquired or persistent compared to the diagnostic biopsy.

### 2.2. Patients

Patients with advanced or non-curative locally advanced NSCLC with no actionable *EGFR* mutations or *ALK* re-arrangements, referred to Department of Clinical Oncology and Palliative Care, Næstved Hospital, were screened for eligibility. Inclusion criteria were: age > 18 years, Eastern Cooperative Oncology Group (ECOG) score of Performance Status (PS) 0–2, measurable disease according to the Response Evaluation Criteria in Solid Tumors, v1.1 (RECIST), ability to understand spoken and written Danish, and written informed consent. Exclusion criteria were: other active cancers and contraindications for systemic treatment.

Clinico-pathological-, treatment- and biopsy-characteristics were collected prospectively. Patients were enrolled from 29 June 2018 to 1 November 2020.

### 2.3. Treatment and Evaluation

According to national treatment guidelines [25], patients were treated with either immunotherapy (pembrolizumab), chemotherapy (carbo- or cis-platin and vinorelbine, or monotherapy vinorelbine), or a combination of immunotherapy and chemotherapy (carbo- or cis-platin, pemetrexed and pembrolizumab, followed by maintenance pemetrexed/pembrolizumab). Of note, the combination treatment was first approved and implemented in the national treatment guidelines in November 2019. The choice of treatment depended on pathology, PD-L1 expression, ECOG PS and renal function/comorbidity status. Immunotherapy was administered until progression, or a maximum of two years, or until unacceptable toxicity. Four to six treatment cycles of chemotherapy were administered, unless progression or unacceptable toxicity was observed. Maintenance treatment with pemetrexed was offered to patients with adenocarcinoma with stable disease/partial response after platinum-doublet therapy and no significant toxicity or decline in PS.

Computed tomography (CT) scans were performed as prescribed after every second or third treatment cycle evaluating the efficacy of the treatment through the RECIST or iRECIST (immunotherapy-related RECIST) criteria.

### 2.4. Biopsy Procedure

All patients had a biopsy performed at diagnosis, and furthermore, a biopsy was performed at time of progressive disease, during or in follow-up after first line treatment. The confirmation of progression by CT-scans and assessment of the location and possible complications of re-biopsy were evaluated at a Multi Disciplinary Team meeting (MDT) with presence of oncologists, pulmonologists, pathologists, and radiologists. The location of biopsy was preferentially performed at the site/sites of progressive disease, determined by CT/PET-CT scans. All re-biopsies were performed at the endoscopy suite at the Department of Respiratory Medicine, Næstved Hospital. A detailed description of the biopsy procedures has been reported previously [24].

### 2.5. DNA Extraction

To confirm the diagnosis and the suitability and representativeness of the material, all histological biopsies/cell blocks were Formalin-Fixed and Paraffin-Embedded (FFPE). The majority of tissue samples (*n* = 79) were prepared from FFPE material. In short, 4–6 sections of 10 µm from each tissue block were used for DNA extraction on the QIACube instrument using the GeneRead DNA FFPE Kit (Qiagen, Hilden, Germany). For patients referred to the Phase 1 Unit at Copenhagen University Hospital, DNA was extracted from fresh tumor tissue and peripheral blood as previously described [9]. In two patients, the tumor was inaccessible for tissue biopsy and tumor profiling was performed using circulating cell-free DNA (cfDNA) purified from blood plasma collected in cell-stabilizing BCT-tubes (Streck Laboratories, La Vista, NV, USA) [26]. Cell-free DNA was extracted from 8 ml plasma using the QIAsymphony Circulating DNA Kit (Qiagen) according to the manufacturer’s instructions using an elution volume of 60 µL. All extracted DNA was quantified using a dsDNA High-Sensitivity (HS) or Broad-Range (BR) assay on a Qubit Fluorometer (Thermo Fisher Scientific, Waltham, MA, USA).

### 2.6. DNA Sequencing and Mutational Analyses

DNA libraries were prepared from a minimum of 10 ng cfDNA or 200 ng FFPE-DNA and hybridized using the TruSight Oncology (TSO) 500 HT gene panel (Illumina) and subsequently, sequenced on the NovaSeq6000 platform to a minimum median coverage of 600×. For samples with a median coverage below 600× due to low tumor cell content or poor tissue quality, data was manually inspected for tumor alterations. For patients referred to the Phase 1 Unit, Whole Exome Sequencing (WES) (*n* = 12) or Whole Genome Sequencing (WGS) (*n* = 4) was performed on fresh tumor tissue and whole blood for germline subtraction. Whole exome library preparation and sequencing has previously been described [9]. For WGS libraries, DNA from either whole blood or tumor was prepared for library using Illumina PCR free Library Preparation, tagmentation kit (Illumina, San Diego, CA, USA) with unique dual indexes. In brief, 100–300 ng DNA was used as input for tagmentation, followed by adaptor ligation and clean-up of the WGS DNA sequencing library. Sequencing was performed as 2 × 150 bp paired-end sequencing on a NovaSeq6000 (Illumina) instrument. DNA-libraries were sequenced with minimum median coverage > 20× and >50× for germline and tumor, respectively. All DNA libraries were quantified and quality-controlled using the appropriate Qubit fluorometer assay and the D5000 ScreenTape Assay for the 4200 TapeStation System (Agilent, Santa Clara, CA, USA).

Sequencing reads were mapped to the hg19/GRCh37 human reference genome using BWA-MEM v0.7.12 software and somatic mutations called using GATK Mutect2 Best Practices guidelines including removal of common polymorphisms present in >5% of the general population (gnomAD). For germline WGS/WES analyses alignment file pre-processing and variant calling was performed by GATK v4.1.0 using Best Practices guidelines. Furthermore, the Illumina TSO500 analysis pipeline was also applied on the cfDNA and FFPE samples to estimate the Tumor Mutational Burden (TMB) and gene amplifications (fold change > 2.2 according to manufactures guidelines).

Mutations were visualized and inspected in QIAGEN Clinical Insight (QCI) Interpret Translational software and each mutation was manually inspected in the sequencing reads using the Integrative Genomics Viewer (IGV). Cut-off values for Mutational Allele Frequencies (MAFs) included MAF ≥ 10% for WES/WGS analyses and MAF ≥ 5% for cfDNA and FFPE. However, if no obvious pathogenic tumor mutations were identified, the MAF filter was removed and the data inspected in QCI for hotspot/well-described tumor alterations down to 1%.

Rare pathogenic or likely pathogenic somatic mutations including nonsense, frameshift, missense, and splice site alterations (+/−2 bp) in cancer-related genes were reported (Appendix A). We included mutations previously identified in cancer tissue (Catalogue of Somatic Mutations in Cancer, COSMIC, https://academic.oup.com/nar/article/47/D1/D941/5146192, accessed on 1 January 2021), relevant databases (e.g., CKB-BOOST, ClinVar), described in the literature, or simply based on the important role of the gene in cancer and the location of the mutation, e.g., not previously reported mutations in the kinase domain of *BRAF*. One molecular biologist interpreted the molecular reports of all patients, blinded to clinical outcome, treatment types, and results of the paired biopsy (diagnostic- or re-biopsy).

### 2.7. Actionable Molecular Alterations and Targeted Treatment

To investigate if the molecular alterations, revealed at progression, could be matched by a targeted treatment in a clinical trial, we searched of the 523 genes defined by the TSO500 HT gene panel in the databases of My Cancer Genome (www.mycancergenome.org, accessed on 1 June 2021) and ClinicalTrials.gov (www.clinicaltrials.gov, accessed on 1 June 2021). In the My Cancer Genome search, we selected only recruiting studies and only malignant solid tumors/NSCLC. In the ClinicalTrials.gov search, we selected only recruiting studies. The search period was June/July 2021. The available in- and exclusion-criteria of every clinical trial were thoroughly investigated to find specifications of the molecular alterations (e.g., transcript and protein variants) and to ensure that our study cohort fulfilled all of the defined criteria. We denoted the molecular specifications, the name and function of the investigational drug, the specified inclusion criteria, ClinicalTrials.gov ID and the official title of the study.

An Actionable Molecular Alteration (AMA) was defined as a molecular alteration with a potential of targeted treatment. For a thorough description of how molecular alterations were called, inspected and reported, see Section 2.6. Targeted treatment was defined as a direct inhibition on the gene level. All phases (Phase I-IV) of studies were included, and with no regards of the evidence level of treatment efficacy. Nevertheless, all AMAs found in our study were searched of in the OncoKB database (http://www.oncokb.org, accessed on 1 July 2021) to investigate any therapeutic level of evidence and FDA approved drugs. OncoKB level of evidence 1 is defined as a “FDA-recognized biomarker predictive of response to an FDA-approved drug in this indication”. OncoKB level of evidence 2 is defined as a “Standard care biomarker recommended by the NCCN or other professional guidelines predictive of response to an FDA-approved drug in this indication”.

The molecular results did not affect second line treatment decisions as clinicians followed the recommendations of the national treatment guidelines. Exceptions were patients referred to the Phase 1 Unit at Copenhagen University Hospital after first line treatment due to limited second line treatment options—e.g., patients receiving both chemotherapy and immunotherapy as first line treatment—or patients with a high chance of benefitting from targeted treatment (e.g., *BRAF* inhibitors).

### 2.8. Statistical Analysis

Descriptive statistics were applied for clinical and pathological characteristics and presented as frequencies, percentages, and median (range). Time to biopsy was calculated as the interval from patient acceptance of biopsy to the performance of biopsy (days). Progression Free Survival (PFS) was defined as the time from first line treatment initiation to radiologically verified progression. Overall Survival (OS) was defined as the time from first line treatment initiation to death of any cause. PFS and OS were calculated by the Kaplan–Meier method through GraphPad prism v8.4.3. Patients with no progression or death by the cut-off date 30 June 2021 were censored. Only patients with a performed re-biopsy were included for further analyses.

## 3. Results

### 3.1. Patient Characteristics

From 29 June 2018 to 1 November 2020 a total number of 254 patients were screened for eligibility of which 150 patients were included in the study. By the cut-off date of 30 June 2021, a total of 119 patients had progression and of these 92 (77%) patients had a re-biopsy performed (Figure 1). A decline in PS was the main reason of not having a re-biopsy performed. Table 1 depicts baseline characteristics and Table 2 depicts patient/biopsy characteristics and second line treatment of patients undergoing re-biopsy (*n* = 92).

### 3.2. Feasibility and Biopsy Characteristics

The median time from patients’ acceptance of re-biopsy to performance of re-biopsy was six days (range, 0–31). The rate of complications to biopsy was 8% (*n* = 7)—including pneumothorax, bleeding, and pneumonia. No severe or life-threatening complications occurred.

The 92 patients underwent 131 biopsy procedures with cytologic biopsy sampling (*n* = 97, 74%) being the most frequent compared to histologic biopsy sampling (*n* = 34, 26%). The most frequent locations of biopsy were lung, lymph nodes, liver, and adrenal glands (Appendix A). In the majority of patients (*n* = 71, 77%), a change in biopsy location from diagnostic to re-biopsy occurred (Figure 2D); however, in only approximately half of patients (*n* = 49%), the change was in between organs (e.g., lung to liver).

A second re-biopsy was performed in seven of the 92 re-biopsied patient cases due to no malignant cells (*n* = 3) or insufficient material for analyses (*n* = 4) in the first re-biopsy. In only one case, no sufficient material was achieved in the second re-biopsy.

### 3.3. Molecular Analyses

All 92 patients had molecular analyses performed of the diagnostic biopsy. At progression, 87 patients (95%) had molecular analyses performed of the re-biopsy. In the remaining five patient cases, no malignant cells (*n* = 3), only suspected malignant cells (*n* = 1) and B-cell lymphoma (*n* = 1) were present in the re-biopsy. This last patient had a biopsy taken from a growing spleen-lesion, revealing B-cell lymphoma. She received systemic treatment for B-cell lymphoma and there were no signs of progression of the NSCLC.

Out of the 92 diagnostic molecular analyses, 91 were performed by TSO500 and one by WES. In the group of re-biopsies, 73 were performed by TSO500, 12 by WES, two had both TSO500 and WES, and four by WGS. WES/WGS were performed if patients were referred to the Phase 1 Unit. Otherwise only TSO500 was performed.

### 3.4. Actionable Molecular Alterations at Re-Biopsy

Overall, 421 molecular alterations in 172 different genes were revealed in the 87 re-biopsies performed at progression. Of the 421 molecular alterations, including mutations, amplifications and fusions, 127 were defined as Actionable Molecular Alterations (AMAs) with the potential of targeted treatment.

In 74 (85%) out of 87 re-biopsied patients at least one AMA was found (Figure 2A). In 37 (50%) of the 74 patients, more than one AMA was revealed (Figure 2B).

Figure 2C illustrates the frequencies of the AMAs. At total of 19 different AMAs were identified, with alterations in *TP53* (*n* = 57, 45%) and *KRAS* (*n* = 31, 24%) being the most frequent, followed by *PIK3CA*, *FGFR1*, *ATM*, *BRAF*, *ERBB2*, *MET*, *AKT2*, *CDK4*, *EGFR*, *MDM2*, *ATR*, *MTOR*, *PDGFRA*, *MEK1*, *CDK6*, *KDR*, and *RET*. A total of 48 different mutations were found in *TP53*, whereas *KRAS* accounted for only 11 different mutations.

The frequency of AMAs was 4 out of 4 patients (100%) treated by combination treatment, 40 out of 43 (93%) in chemotherapy-treated patients and 30 out of 40 (75%) in immunotherapy-treated patients.

Figure 3 illustrates the frequencies and protein/transcript variants (top) of the different mutations (A) and the frequencies of the different amplifications (B). The number of recruiting studies for each AMA (middle) and the distribution of the different phases (Phase I–IV) of studies (bottom) available are illustrated. Only studies including the specific protein/transcript variant of AMAs found in our study were denoted. FDA-approved drugs in NSCLC with OncoKB therapeutic level of evidence 1 or 2 are listed. Appendix A specifies the name of the study drugs and ClinicalTrials.gov IDs of the mutations (1) and amplifications (2). The AMAs with most recruiting studies were *ERBB2* mutations (*n* = 37), *MET*-amplifications (*n* = 22), and *BRAF* mutations (*n* = 17). *RET*-fusions were the only AMA with recruiting Phase IV studies.

### 3.5. Acquired and Persistent Actionable Molecular Alterations

A total of 222 (53%) of the 421 different molecular alterations revealed at re-biopsy, were acquired—i.e., not identified in the diagnostic biopsy. In addition, 125 molecular alterations were only present at the diagnostic biopsy (Figure 4B). Overall, the most frequent acquired molecular alterations were noted in *TP53* (*n* = 16), but in general heterogeneity was observed (Figure 4C).

In 66 (76%) of the 87 re-biopsied patients, an acquired molecular alteration was present with a median of two (1–14) acquired molecular alterations (Figure 4A). Acquired molecular alterations were observed with almost equal frequencies in immunotherapy-treated patients (30 out of 40, 75%) and chemotherapy-treated patients (33 out of 43, 77%) (Figure 4D). We observed a wider heterogeneity in the acquired molecular alterations in patients treated by chemotherapy, involving 93 different genes, compared to only 70 different genes in patients treated by immunotherapy.

Looking specifically at AMAs, an acquired alteration was seen in 36 (28%) of the 127 AMAs detected at progression (Figure 2A, top). By performing a thorough search of these specific acquired alterations from the results of the diagnostic biopsy, we found that only 15 alterations (12%) were truly acquired. In general, these alterations were found in very low frequencies in the diagnostic samples with a possible low fraction of tumor cells.

A total of 24 (32%) of the 74 patients had acquired AMAs, which corresponds to 28% of all of the 87 re-biopsied patients (Figure 2A, top). Of these, 11 were chemotherapy-treated, 11 were immunotherapy-treated, and two had combination treatment. Figure 3A,B illustrates the distribution of acquired and persistent alterations in the AMAs.

### 3.6. Second Line Treatment, Median Overall Survival and Performance Status

A total of five patients (5%) of the 92 re-biopsied patients received targeted treatment as second or further lines of treatment based on the AMAs revealed at re-biopsy (Figure 5B and Figure 2D) with a median OS of approximately 23 months (702 days, range 310–789). Patients receiving non-targeted treatment and no systemic treatment had a median OS of approximately 18 months (548 days, range 74–1023) and 10 months (290 days, range 64–1029), respectively.

More than one third (*n* = 35, 38%) did not receive any systemic second line treatment—including three patients in continued post-treatment observation after stereotactic thoracic irradiation (*n* = 1), calcium electroporation of subcutaneous metastases [27] and thoracic irradiation (*n* = 1) and stereotactic irradiation of liver-metastasis (*n* = 1). The main reason for not receiving systemic second line treatment was patients’ deterioration, illustrated in Figure 5A by a decline in PS (*n* = 18, 60%). Overall, a change in PS was observed in 52 (57%) of the 92 re-biopsied patients with 46 (50%) patients experiencing an increase/worsening of PS.

## 4. Discussion

The potential gain of re-biopsy in advanced NSCLC is not readily determined from current data, as re-biopsy is not integrated in current treatment-guidelines. Most knowledge comes from precision medicine trials, reporting from cohorts of patients with a variety of different cancer types, receiving different kinds of study treatments. Patients are highly selected, fulfilling the inclusion criteria of the specific trial, and often with exhaustion of standard treatment options.

Our study reports comprehensive molecular profiling data of a real-time cohort of 150 advanced or non-curative locally advanced NSCLC patients, having the availability of a re-biopsy on progression after first line treatment.

### 4.1. Actionable Molecular Alterations at Re-Biopsy

To evaluate the gain of re-biopsy, we focused on AMAs at re-biopsy and found that the majority of patients (85%) had at least one AMA with the potential of targeted treatment (Figure 2A). Presley et al. found in their retrospective study including 875 stage IIIB/IV or un-resectable non-squamous NSCLC patients with performance of broad-based genomic sequencing (>30 genes), a similar frequency of 88.9% [6]. Lower and more varying frequencies of 40–75% have been reported from precision medicine trials [9,10,11,12,13,14,15,16,17,18,19]. The technological advances in molecular profiling over time have contributed to high-throughput analyses, which can explain some of the variance over time. Additionally, the increasing number of available targeted treatments also contributes to define more molecular alterations as actionable. The possible higher frequency of AMAs in advanced NSCLC patients could indicate that these patients tend to have more molecular alterations, which corresponds to a general higher tumor mutational burden compared to other cancer types [28].

### 4.2. Targeted Treatment Options

In our study, we found that *TP53* and *KRAS* mutations were the most frequent AMAs (Figure 2B), which corresponds to the findings by Presley et al. [6] and by Réda et al.’s [15] NSCLC cohorts. *TP53*’s categorization as an AMA could be debated, as no prior studies have showed promising results by targeting *TP53*. Based on the Phase I/II study (NCT04383938) of Eprenetapopt (APR-246), a first-in-class mutant p53 re-activator, *TP53* was included as an AMA. Categorization of *PIK3CA* amplifications and *ATM* mutations as AMAs could also be debated, as Gedatolisib (NCT03065062) and M4076 (NCT04882917), respectively, are the only available targeted treatments.

The number of available studies offering targeted treatments (Figure 3A,B) did not reflect the frequencies of the different alterations found in our study. Many of the less frequent alterations, like *RET* fusions and *MET* amplifications, were represented in many studies, in line with the growing experience of very beneficial response rates when targeting these low-frequent fusions/amplifications.

Another important aspect is alterations present in different cancer types with the potential of favorable responses in many patients, illustrated in our study by a large number of studies of *ERBB2* and *BRAF* inhibitors.

### 4.3. The Possibility and Availability of Targeted Treatment

The responses to targeted therapy has increased from approximately 3% in 2006 to 7% in 2020 [4], which gives hope to the field of precision medicine. Precision medicine trials reports of only 8–27% actually receiving targeted treatment [9,10,11,12,13,14,15,16,17,18,19] and response rates of 0.8–3% [29].

In the cohort described here, only 5% of patients received targeted treatment in any treatment line (Figure 5B), which corresponds to the 4.5% of non-curative NSCLC patients, reported by Presley et al. [6]. The lower frequencies compared to precision medicine trials is probably caused by our study design, reflecting a “real-life” cohort. In addition, the patient cohort only received first line treatment before comprehensive molecular profiling, whereby the choice of second line treatment relied on national treatment guidelines [25].

Another important aspect is the availability of targeted treatments. Only 50% of the suggested targeted treatment options found in our study, were available nationwide. Despite a public health-care system and free transition to departments with available protocolled targeted treatment, few patients had the opportunity of actually receiving the proposed targeted treatment.

In precision medicine trials, one of the primary reasons of not receiving the intended targeted treatment is patients’ deterioration [9,10,11,12,13,14,17,18]. In our study, more than one third of patients (38%) did not receive any systemic second line treatment, primarily due to a decline in PS, observed in 60% of patients (Figure 5A). Presley et al. found a higher proportion of almost 53% not receiving second line treatment, probably illustrating that patients with non-curative NSCLC, in general, is a fragile patient cohort.

### 4.4. The Necessity of Re-Biopsy

Only few studies have focused on the significance of a re-biopsy in terms of novel potential targets and potential resistance mechanisms—compared to the findings at the diagnostic biopsy [21,22,30]. A recent study by Joris Van de Haar et al [21], observed 94% concordance in Whole Genome Sequencing of paired biopsies, when focusing on biomarkers for clinical trial enrolment, in a heterogenous cohort of 213 metastatic cancer patients receiving a variety of different treatments. Accordingly, evidence of the optimal time-point of comprehensive molecular profiling is still lacking and the debate of the need of a re-biopsy at progression to account for therapeutic pressure and tumor heterogeneity is on-going [20].

We found that more than half (53%) of the molecular alterations at re-biopsy were acquired—potentially representing therapeutic pressure. However, no clear tendencies or groupings of alterations were observed (Figure 4C).

To clarify if a re-biopsy at progression contributes with important molecular information in treatment-guidance, we looked specifically at the AMAs and found that approximately 30% were acquired (Figure 3A,B), mainly *PIK3CA* amplifications/mutations and *FGFR1* amplifications.

It is important to note, that some of the possible acquired molecular alterations could represent the known inter-tumoral heterogeneity [31,32,33,34] as 77% had the re-biopsy taken from another location than the diagnostic biopsy (Figure 2D). In addition, with thorough investigation of the diagnostic biopsies, some of the possible acquired alterations could be revealed, potentially reflecting a higher concordance and/or diagnostic biopsies with a low tumor fraction.

### 4.5. The Clinical Relevance of Re-Biopsy

Many aspects are important, when debating the clinical relevance of re-biopsy at progression in advanced NSCLC. Based on the findings in our study where only 62% received second line systemic treatment due to a decline in PS, it is important to be aware of the patient’s capability of receiving systemic treatment. Despite the finding of AMAs in the majority of patients (85%), it is essential to investigate the availability of targeted treatments, as in our study only 50% of the suggested targeted treatment was available nationwide. Finally, it is important to include the risk of missing potential AMAs by not performing a re-biopsy, as we found that approximately one third of the AMA’s were only detected at re-biopsy.

In a setting without economic limitations, a comprehensive molecular profiling at diagnosis and at progression are preferable. In most countries, where comprehensive molecular profiling is not included in the standard analyses at diagnosis, a re-biopsy at progression after first line immunotherapy/combination treatment could, in patients with acceptable performance status, based on our findings, be a recommended course to (i) avoid redundant and expensive profiling in patients not capable of receiving further treatment (ii) to increase the possibility of catching all the existing AMAs. It requires a suitable logistic set-up for the biopsy-procedure and laboratory facilities to have results of analyses within an acceptable time frame.

### 4.6. Limitations and Perspectives

This study has several limitations that impact conclusions on clinical utility of re-biopsy. We did not include the impact of Tumor Mutational Burden (TMB), Micro Satellite Instability (MSI) or fusions (only in WES/WGS), which could have increased the number of patients amenable to a protocolled treatment. In addition, we did not include patients in the AMA-category if they had a potential of semi-targeted treatment (mechanistic inhibition of the molecular alteration) or non-targeted treatment (the molecular alteration was an inclusion criteria). As discussed, we defined *TP53* as an AMA due to one recruiting study. This categorization could, as mentioned, be debated. The AMAs were based upon all the available protocolled targeted treatments worldwide (Clinical.trials.gov and MyCancerGenome). Clinical aspects and resistance mechanisms were not included in the assessment of whether an AMA could be targeted or not. Furthermore, some of the acquired alterations could represent inter-tumoral heterogeneity as 77% of patients had the re-biopsy taken from another location than the diagnostic biopsy.

## 5. Conclusions

Our study revealed AMAs with the potential of targeted treatment in 85% of patients. Approximately 30% of the AMAs were potentially acquired. Only 62% of patients were capable of actually receiving further lines of systemic treatment due to a decline in PS. Just 5% received targeted treatment—including *ERBB2*-, *EGFR*-, *BRAF*- and *RET*-directed treatment.

## Figures and Tables

**Figure 1 cancers-14-00132-f001:**
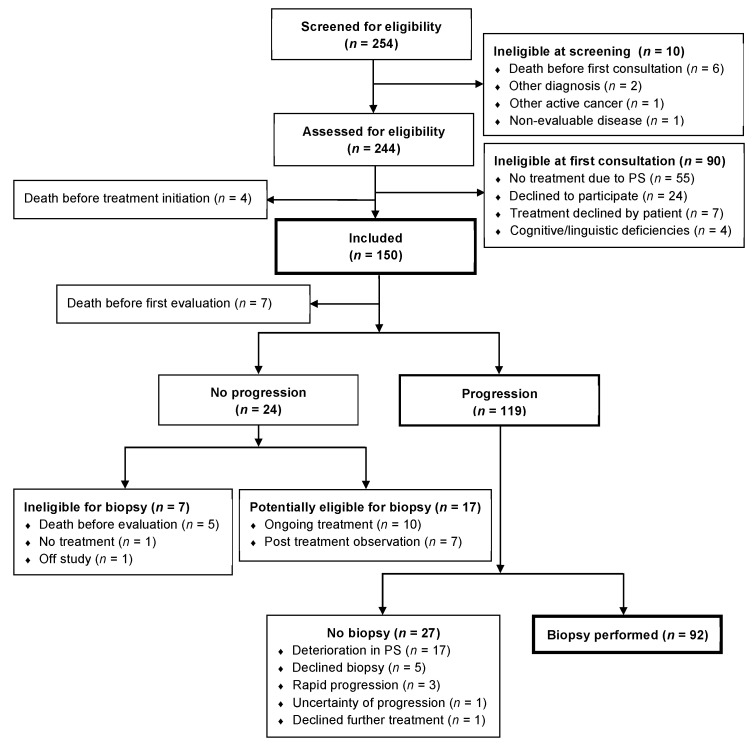
Consort diagram.

**Figure 2 cancers-14-00132-f002:**
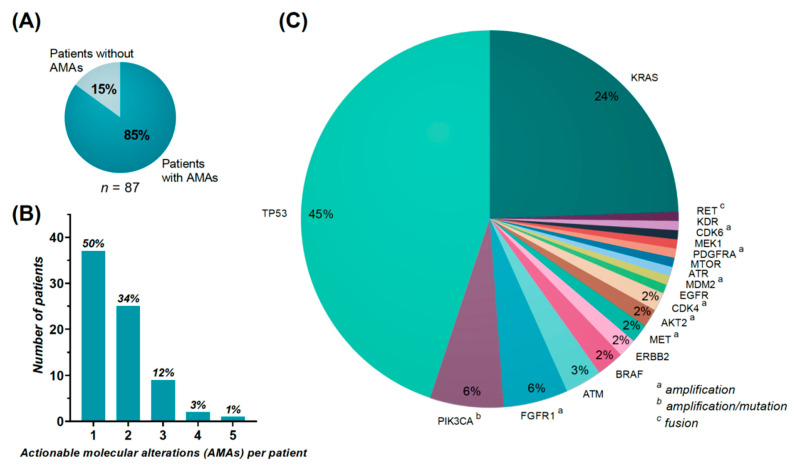
(**A**) illustrates the distribution of patients with or without Actionable Molecular Alterations (AMAs). (**B**) illustrates the number of AMAs per patient. (**C**) shows the frequencies of the AMAs. (**D**) illustrates for each patient (columns): Histology of the diagnostic biopsy, occurrence of location changes between the diagnostic biopsy and re-biopsy, molecular alterations, and type of second line treatment.

**Figure 3 cancers-14-00132-f003:**
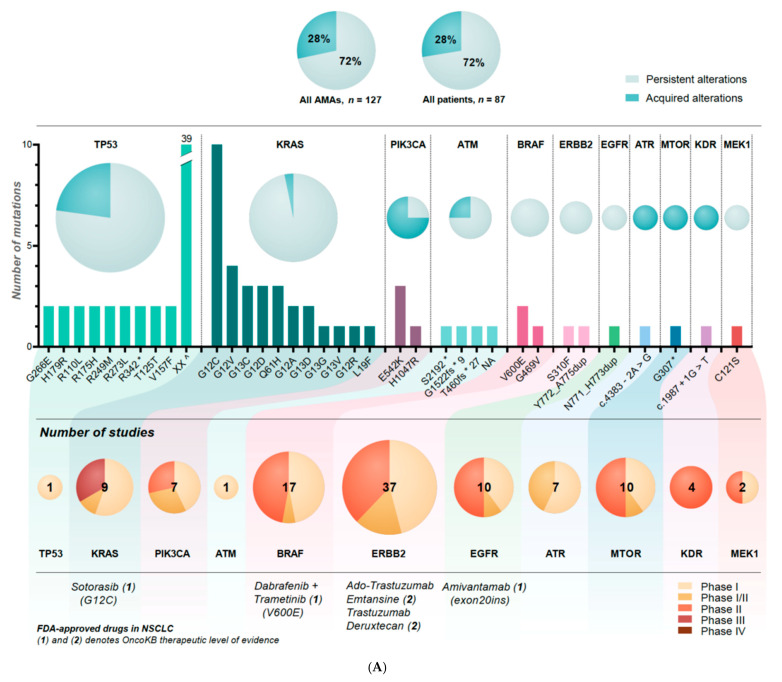
(**A**,**B**) Top illustrates the distribution of persistent and acquired actionable molecular alterations (AMAs) in all the AMAs and in all the re-biopsied patients. Middle shows the frequencies and protein/transcript variants of the different mutations (**A**)/amplifications (**B**) including the distribution of persistent and acquired mutations. ^c^ denotes fusion. XX ^ represents different alterations occurring one time. * denotes stop codon. Bottom illustrates the corresponding number of recruiting studies available including the respective phases and FDA-approved drugs in NSCLC.

**Figure 4 cancers-14-00132-f004:**
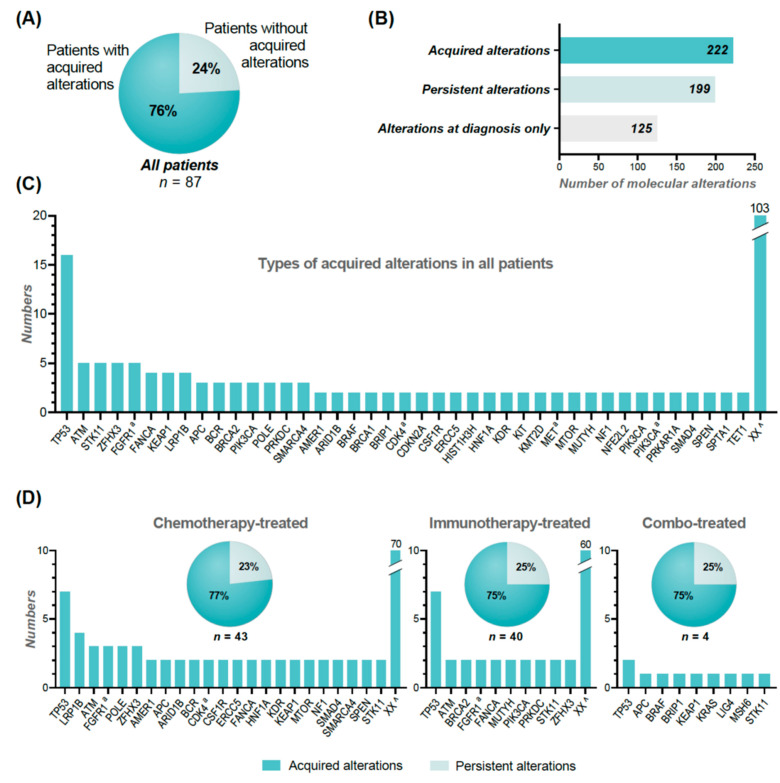
(**A**) illustrates the percentage of patients with or without acquired alterations. (**B**) shows the number of acquired, persistent, and diagnostic (only) alterations. (**C**) illustrates the types and frequencies of the acquired alterations in all patients. (**D**) illustrates the types and frequencies of acquired alterations in chemotherapy-treated, immunotherapy-treated, and combo-treated patients. ^a^ denotes amplifications. XX ^ represents different alterations occurring one time.

**Figure 5 cancers-14-00132-f005:**
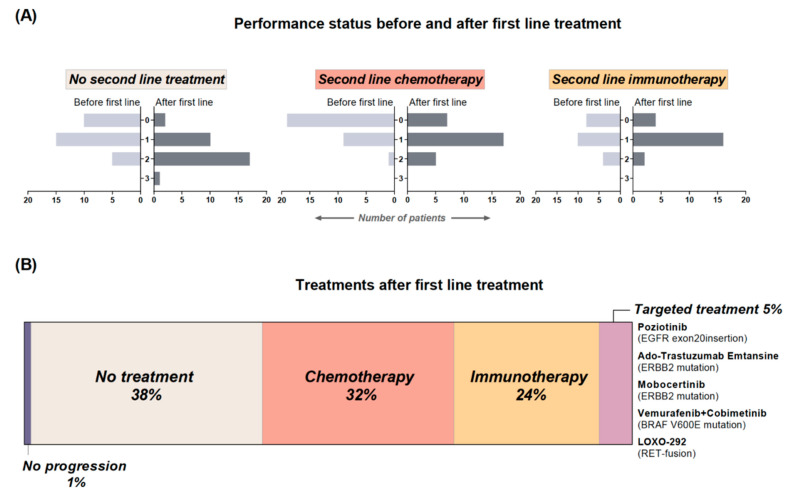
(**A**) illustrates Performance Status (PS) before and after first line treatment in patients without further systemic treatment, chemotherapy-treated, and immunotherapy-treated as second line treatment. (**B**) shows the distribution of patients according to treatment after first line.

**Table 1 cancers-14-00132-t001:** Baseline characteristics of re-biopsied patients.

Baseline Characteristics of Re-Biopsied Patients (*n* = 92)		
**Sex**	***n* (%)**	
Male	49 (53.2)	
Female	43 (46.7)	
**Age**	**Median Years (Range)**	
	67 (45–84)	
**ECOG Performance Score at Diagnosis**	** *n* ** **(%)**	
PS 0	43 (46.7)	
PS 1	39 (42.4)	
PS 2	10 (10.9)	
PS 3	0 (0.0)	
**Histology**	** *n* ** **(%)**	
Adenocarcinoma	67 (72.8)	
Squamous Cell Carcinoma	21 (22.8)	
Not Otherwise Specified	4 (4.3)	
**Stage, IASCL, 8th Edition**	** *n* ** **(%)**	
IIIA	1 (1.0)	
IIIB	6 (6.5)	
IIIC	5 (5.4)	
IVA	47 (51.1)	
IVB	33 (35.9)	
**PD-L1 Expression**	** *n* ** **(%)**	
<1%	28 (30.4)	
≥1% <50%	15 (16.3)	
≥50%	48 (52.2)	
Unknown	1 (1.1)	
**Treatment Type**	** *n* ** **(%)**	**No of Cycles (Range)**
Carbo- or cis-platin/vinorelbine ^1^	42 (45.7)	4 (1–10)
Vinorelbine	2 (2.2)	4 (2–6)
Pembrolizumab	44 (47.8)	6 (2–32)
Caboplatin/pemetrexed/pembrolizumab	4 (4.3)	8 (6–16)

^1^ With/without maintenance pemetrexed.

**Table 2 cancers-14-00132-t002:** Patient and biopsy characteristics at time of progression and second line treatment.

Characteristics at Time of Progression and Second Line Treatment (*n* = 92)	
**Median PFS**	**Days (Range)**
All patients (*n* = 92) ^1^	137 (23–805+)
Chemotherapy treated (*n* = 44)	89 (23–311)
Immunotherapy treated (*n* = 48) ^1,3^	174 (26–805+)
**ECOG Performance Score at Progression**	** *n* ** **(%)**
PS 0	17 (18.5)
PS 1	47 (51.1)
PS 2	27 (29.3)
PS 3	1 (1.1)
**Biopsy Locations**	** *n* ** **(%)**
Lung	40 (43.5)
Lymph nodes	24 (26.1)
Liver	8 (8.7)
Adrenal glands	6 (6.5)
Pleura	4 (4.3)
Pleural fluid	2 (2.2)
Subcutaneous metastases	2 (2.2)
Spleen	2 (2.2)
Bone	2 (2.2)
Ascites	1 (1.1)
Brain	1 (1.1)
**Second Line Treatment**	** *n* ** **(%)**
No treatment	35 (38.0)
Chemotherapy	29 (31.5)
Immunotherapy ^2^	22 (23.9)
Targeted treatment	5 (5.4)
B-cell lymphoma treatment ^1^	1 (1.1)

^1^ Including one patient without progression as the re-biopsy revealed B-cell lymphoma. ^2^ Including four patients continuing immunotherapy beyond progression. ^3^ Including combination treatment.

## Data Availability

The data presented in this study are available in Appendix A.

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
