# Peer review of "Actionable Molecular Alterations Are Revealed in Majority of Advanced Non-Small Cell Lung Cancer Patients by Genomic Tumor Profiling at Progression after First Line Treatment"

_cancers, 2021, doi:10.3390/cancers14010132_

Round 1

Reviewer 1 Report

The authors described the molecular alterations during systemic therapies including chemotherapy, or immunotherapy in patients with NSCLC absent for EGFR/ALK, exploring the significance of comprehensive genomic profiling at disease progression in this populations.

I have some minor comments as below.

1. I am wondering if the definition of actionable molecular alterations (AMA) is acceptable. For examples, TP53 alterations are usually passanger gene alterations.

2.  The author may need to discuss the reason why very few patients received Pemetrexed-based chemotherapies, despite most of patients were Adenocarcinoma.

3. Some patients were supposed to have AMAs in multi-genes. For readers to understand which genes co-alters in one patient, I recommend the authors to make Figure1C like Oncoprint.

4.  Which site did you perform biopsy at diagnosis/progression?

If these are different each other, the analysis may just see tumors heterogeneity, not acquired alterations. The author may need to discuss these limitation of this study.

Author Response

We thank the Editor and reviewers for their time and constructive comments. We have thoroughly revised the manuscript with regard to these comments.

Please find below the reviewers comments followed by an account of steps taken to change the manuscript accordingly. We have indicated the changes by listing the paragraph, page and line number to any action taken as well as including a revised manuscript with track-changes.

We have revised Figure 5 and added Figure 2D, as suggested by Reviewer #1

Reviewer #1

Point 1

Reviewer comment: I am wondering if the definition of actionable molecular alterations (AMA) is acceptable. For examples, TP53 alterations are usually passenger gene alterations.

Author reply: Indeed this could be debated as mentioned in the manuscript:

Page 15, Section: 4.2 Targeted treatment options, line 3:

“TP53’s categorization as an AMA could be debated, as no prior studies have showed promising results by targeting TP53. Based on the Phase I/II study (NCT04383938) of Eprenetapopt (APR-246), a first-in-class mutant p53 re-activator, TP53 was included as an AMA.”

Action taken:

We have listed this as a limitation with the following sentence on page 17 in the section 4.6 Limitations and perspectives, line 7 “As discussed, we defined TP53 as an AMA due to one recruiting study. This categorization could, as mentioned, be debated.”

Point 2

Reviewer comment: The author may need to discuss the reason why very few patients received Pemetrexed-based chemotherapies, despite most of patients were Adenocarcinoma.

Author reply: We thank the reviewer for the relevant comment. Combination treatment with carboplatin, pemetrexed and pembrolizumab was only given to four patients as this treatment regimen was first approved on the 5th of November 2019 in Denmark to patients with NSCLC, adenocarcinoma, PD-L1 expression ³ 1 % < 50% and Performance Status 0-1. Inclusion of patients was initiated on the 29th of June 2018 and ended on the 1st of November 2020.

Action taken: We have added a sentence on Page 3, Section 2.3 Treatment and evaluation, line 5: “Of note, the combination treatment was first approved and implemented in the national treatment guidelines by November 2019.”

Point 3

Reviewer comment: Some patients were supposed to have AMAs in multi-genes. For readers to understand which genes co-alters in one patient, I recommend the authors to make Figure1C like Oncoprint.

Author reply: Thank you for the very useful recommendation to clarify data for the readers. 

Action taken: We have added Figure 2D illustrating for each patient (columns): the histology, the occurrence of location changes between the diagnostic- and re-biopsy, the actionable molecular alterations, and type of second line treatment.

Point 4

Reviewer comment: Which site did you perform biopsy at diagnosis/progression? If these are different each other, the analysis may just see tumors heterogeneity, not acquired alterations. The author may need to discuss these limitations of this study.

Author reply: We thank reviewer for the comment and the highly relevant question.

We have been aware of this aspect and have commented on this in the following sections:

Page 8, Section: 3.2 Feasibility and biopsy characteristics, line 8

”In the majority of patients (n=71, 77%), a change in biopsy location from diagnostic to re-biopsy occurred, however in only approximately half of patients (n=48, 52%) the change was in between organs (e.g. lung to liver).”

Page 16, Section: 4.4 The necessity of re-biopsy, line 17

”It is important to note, that some of the possible acquired molecular alterations could represent the known inter-tumoral heterogeneity [31-34] as 77% had the re-biopsy taken from another location than the diagnostic biopsy.

We have previously published a paper focusing on sites of biopsy at diagnosis and progression in this cohort and the possible impact of heterogeneity of the results. For further details see reference:

Frank, M.S.; Bødtger, U.; Høegholm, A.; Stamp, I.M.; Gehl, J. Re-biopsy after first line treatment in advanced NSCLC can reveal changes in PD-L1 expression. Lung cancer (Amsterdam, Netherlands) 2020, 149, 23-32, doi:10.1016/j.lungcan.2020.08.020.

Action taken:

1) We have on page 17 in the section 4.6 Limitations and perspectives, line 11 included the following sentence: “Furthermore, some of the acquired alterations could represent inter-tumoral heterogeneity as 77% of patients had the re-biopsy taken from another location than the diagnostic biopsy.”

2) We have added Figure 2D to show the occurrence of location changes. 

Reviewer 2 Report

The authors evaluated the gain in actionable molecular targets at time of progression compared to time of diagnosis. I want  to congratulate the authors. It is an interesting and well presented paper. I only have minor remarks.

Minor:

  1. L.200. In the method section: "..(AMA) was defined as a molecular alteration with a potential target treatment." For clarity, please add precisely here how molecular alterations were defined based on the NGS data (e.g. which thresholds were applied).
  2. "Naestved hospital" change to "Næstved hospital"

Author Response

We thank the Editor and reviewers for their time and constructive comments. We have thoroughly revised the manuscript with regard to these comments.

Please find below the reviewers comments followed by an account of steps taken to change the manuscript accordingly. We have indicated the changes by listing the paragraph, page and line number to any action taken as well as including a revised manuscript with track-changes.

We have revised Figure 5 and added Figure 2D, as suggested by Reviewer #1

Reviewer #2

Point 1

L.200. In the method section: "..(AMA) was defined as a molecular alteration with a potential target treatment." For clarity, please add precisely here how molecular alterations were defined based on the NGS data (e.g. which thresholds were applied).

Author reply: We thank the reviewer for the positive comments on the study. Regarding the definition of molecular alterations, we have made a thorough description on Page 4, Section 2.6 DNA Sequencing and mutational analyses, line 28:

Cut-off values for mutational allele frequencies (MAFs) included MAF ≥10% for WES/WGS analyses and MAF ≥5% for cfDNA and FFPE. However, if no obvious pathogenic tumor mutations were identified, the MAF filter was removed and the data inspected in QCI for hotspot/well-described tumor alterations down to 1%. Rare pathogenic or likely pathogenic somatic mutations including nonsense, frameshift, missense, and splice site alterations (+/- 2bp) in cancer-related genes were reported (Suppl. Table 1). We included mutations previously identified in cancer tissue (Catalogue of Somatic Mutations in Cancer, COSMIC), relevant databases (e.g. CKB-BOOST, ClinVar), described in the literature, or simply based on the important role of the gene in cancer and the location of the mutation e.g. not previously reported mutations in the kinase domain of BRAF.”

Action taken: To clarify these elements, we have added a sentence on Page 5, Section 2.7 Actionable molecular alterations and targeted treatment, line 14: For a thorough description of how molecular alterations were called, inspected and reported, see section 2.6.”

Point 2

"Naestved hospital" change to "Næstved hospital"

Author reply: We thank the reviewer for the kind notice.

Action taken: We have corrected this on

Page 2, Section 2.1 Study design, line 3: “The study is a prospective, explorative single-centre study (ClinicalTrials.gov NCT03512847) conducted at Department of Clinical Oncology and Palliative Care, Zealand University Hospital (Næstved Hospital).”

Page 3, Section 2.2 Patients, line 3:

“Patients with advanced or non-curative locally advanced NSCLC with no actionable EGFR mutations or ALK re-arrangements, referred to Department of Clinical Oncology and Palliative Care, Næstved Hospital…”